# Inhibitory Effect of *Lactococcus* and *Enterococcus faecalis* on Citrobacter Colitis in Mice

**DOI:** 10.3390/microorganisms12040730

**Published:** 2024-04-03

**Authors:** Ullah Naveed, Chenxi Jiang, Qingsong Yan, Yupeng Wu, Jinhui Zhao, Bowen Zhang, Junhong Xing, Tianming Niu, Chunwei Shi, Chunfeng Wang

**Affiliations:** 1College of Veterinary Medicine, Jilin Agricultural University, Changchun 130118, China13087175771@163.com (C.J.); 15684210257@163.com (Y.W.); gaokuipeng@163.com (J.Z.); 18686401118@163.com (T.N.); 2Jilin Provincial Engineering Research Center of Animal Probiotics, Jilin Provincial Key Laboratory of Animal Microecology and Healthy Breeding, Jilin Agricultural University, Changchun 130118, China; 3Key Laboratory of Animal Production and Product Quality Safety of Ministry of Education, Jilin Agricultural University, Changchun 130118, China

**Keywords:** *Lactobacillus* combination, *Citrobacter rodentium*, cellular immunity, intestinal microbiota

## Abstract

Probiotics are beneficial for intestinal diseases. Research shows that probiotics can regulate intestinal microbiota and alleviate inflammation. Little research has been done on the effects of probiotics on colitis in mice. The purpose of this study was to investigate the inhibitory effect of the strains isolated and screened from the feces of healthy piglets on the enteritis of rocitrobacter. The compound ratio of isolated *Lactobacillus* L9 and *Enterococcus faecalis* L16 was determined, and the optimal compound ratio was selected according to acid production tests and bacteriostatic tests in vitro. The results showed that when the ratio of *Lactobacillus* L9 to *Enterococcus faecalis* L16 was 4:1, the pH value was the lowest, and the antibacterial diameter was the largest. Then, in animal experiments, flow cytometry was used to detect the number of T lymphocytes in the spleen and mesenteric lymph nodes of mice immunized with complex lactic acid bacteria. The results showed that the number of T lymphocytes in the spleen and mesenteric lymph nodes of mice immunized with complex lactic acid bacteria significantly increased, which could improve the cellular immunity of mice. The microbiota in mouse feces were sequenced and analyzed, and the results showed that compound lactic acid bacteria could increase the diversity of mouse microbiota. It stabilized the intestinal microbiota structure of mice and resisted the damage of pathogenic bacteria. The combination of lactic acid bacteria was determined to inhibit the intestinal colitis induced by Citrobacter, improve the cellular immune response of the body, and promote the growth of animals.

## 1. Introduction

As probiotics are active microorganisms that can have a positive impact on the body, they can colonize the body, regulate the intestinal microbiota and inhibit the growth and reproduction of harmful bacteria. Therefore, probiotics have important research significance as a substitute for antibiotics [1]. In the study of probiotics, *Lactobacillus* has always been the focus of research. Studies have shown that the combined application of multiple lactobacilli is stronger than the effect of a single *Lactobacillus*. Adding a *Lactobacillus* combination to the diet of weaned piglets can effectively improve the intestinal environment and facilitate the absorption of nutrients [2].

*Citrobacter. rodentium* (*C. rodentium* ATCC51459) is a pathogenic bacterium that can colonize the colon of mice and cause attaching and effacing (A/E) injury [3]. Studies have shown that 10^8^–10^9^ CFU of orally administered *C. rodentium* can be observed on the surface of caecal lymphoid tissue several hours later. *C. rodentium* adapts to environmental conditions in the gastrointestinal tract and promotes its own colonization in colon tissues by effector proteins encoded by its virulence islands [4]. This activity is similar to the pathogenic mechanism of enteropathogenic *Escherichia coli* (EPEC) and enterohemorrhagic *Escherichia coli* (EHEC), both of which use the A/E mechanism to bind tightly to the host colon. *C. rodentium* adapts to environmental conditions in the gastrointestinal tract and promotes colonization in colon tissues by effector proteins encoded by its virulence islands. A/E injury plays an important role in the induction of colitis by *C. rodentium*. Therefore, *Citrobacter rodentium* infection is often used as a model to study colitis development [5].

In this experiment, the biological characteristics and safety evaluation of *Lactobacillus* isolated from the feces of piglets were analyzed, and two dominant *Lactobacillus* strains, L9 and L16, were screened. Then, Lactobacilli L9 and L16 were mixed in a 4:1 ratio. After the optimal ratio was selected, the mice were treated intragastrically to observe the growth-promoting effect of the *Lactobacillus* combination on the average daily gain, average daily feed intake and feed/gain of mice. Then, *C. rodentium* was administered intragastrically, and the mouse weight and number of *C. rodentium* in feces were observed. Immunoassays of IFN-γ and IL-10 were used to evaluate the inhibitory effect of the *Lactobacillus* combination on *Citrobacter*-induced colitis in vivo.

## 2. Materials and Methods

### 2.1. Test Strain

Porcine *Lactobacillus* L9 and *Enterococcus faecalis* L16 were isolated and screened from the feces of piglets by the Laboratory of Jilin Animal Probiotics Engineering Research Center, Jilin Agricultural University, and then identified by 16S rDNA sequencing by Kumei Biotechnology Co., Ltd. (Changchun, China). The sequencing results were compared by Blast in NCBI. L9 was identified as Micrococcus lactis, and its gene number was GenBank: AJ249891.1. L16 is Enterococcus faecalis, its GenBank: MW349975.1. Then, the growth curve and acid production curve of the two strains were measured, their stress resistance, adhesion and antibacterial ability were analyzed, and their safety was evaluated. Finally, the two strains could be used as functional strains in microecological preparations.

*C. rodentium* was donated by Dr. Haining Shi, Harvard Medical School, USA, and preserved by Jilin Provincial Animal Probiotics Engineering Research Center, Jilin Agricultural University.

### 2.2. Animals and Ethics Statement

Female BALB/c mice aged 3–4 weeks were purchased from Beijing Huafukang Biotechnology Co., Ltd. (Beijing, China). The mice were allowed to drink and eat freely and received no antibiotics. The whole animal experiment met the requirements of the Animal Management and Ethics Committee of Jilin Agricultural University (protocol number JLAU08201811).

### 2.3. Optimum Compound Ratio of the Strains

After *Lactobacillus* L9 and *Enterococcus faecalis* L16 were activated three times, they were transferred to culture for 24 h. The concentration of viable bacteria was adjusted to 1.0 × 10^9^ CFU/mL, with compounded according to the ratio of 1:1, 1:2, 1:3, 1:4, 2:1, 3:1, and 4:1 of porcine *Lactobacillus lactis* L9 and porcine *Enterococcus faecalis* L16, respectively, which were transferred to MRS broth medium; a single strain was established as the control.

The cultures were incubated in a constant-temperature incubator (Thermo Field, Boston, MA, USA) at 37 °C for 24 h. Then, the pH of each compound bacterial solution and single bacterial solution was measured with a pH meter, and the experiment was repeated three times.

### 2.4. Bacteriostatic Test

First, pathogenic bacteria were inoculated onto the surface of LB’s solid medium, and then a hole punch was used to make 4 holes in the solid medium. Sterile water was added to the first hole as the control hole, L9 was added to the second hole, L16 was added to the third hole, and L9 + L16 was added to the fourth hole according to the ratio of composition (7 groups in total according to different mixing ratios). The size of the inhibition ring was observed after 8 h culture.

### 2.5. Grouping of Experimental Animals

Twenty-four female BALB/C mice aged 3–4 weeks were randomly divided into the PBS group and compound *Lactobacillus* group (compound ratio: L9:L16 = 4:1), with 12 mice in each group. The mice were fed basic feed and had free access to drinking water in an SPF animal house.

The PBS group was given 200 µL PBS 14 days before the experiment, and the compound *Lactobacillus* group was given 200 µL live *Lactobacillus* at a concentration of 1 × 10^9^ CFU/mL 14 days before the experiment. The body weight feed intake of the mice was recorded every other day. The PBS group and the compound *Lactobacillus* group were given 200 µL *C. rodentium* at a concentration of 1 × 10^9^ CFU/mL for two days starting on the 15th day of the experiment. The body weight was recorded every other day after *C. rodentium* intragastric administration, and feces were collected every other day. Blood samples were collected from the eyes of mice in the two experimental groups on the 14th day after the administration of *Lactobacillus* and the 14th day after the challenge. Mice were sacrificed 14 days after the challenge; the duodenum and colon of the PBS group and compound *Lactobacillus* group were soaked in paraformaldehyde for fixation; and paraffin pathological sections were made by HE staining.

Effects of compound strains on the average daily weight gain of mice

During the period of *Lactobacillus* combination administration to mice by gavage, the mice were weighed every two days, and the weight of the mice in the PBS group was used as the control. The data were recorded, the average daily gain (ADG) was calculated, and a chart was drawn.

### 2.6. Effects of Compound Strains on the Average Daily Feed Intake of Mice

During the period of *Lactobacillus* combination administration to mice by gavage, the feed intake of piglets was recorded every two days and compared with that of mice in the PBS group, and a chart was drawn after recording and calculating the average daily feed intake (ADFI).

### 2.7. Effects of Compound Strains on the Feed/Gain Ratio

The feed-to-gain ratio can directly reflect the feed conversion rate and affect the economic benefits of the breeding industry. Therefore, the calculation of the ratio of feed to gain is particularly important, and the calculation formula is as follows: ratio of feed to gain = average daily feed intake (ADFI)/average daily gain (ADG).

### 2.8. The Total Antioxidant Capacity of Serum Determined by ELISA

On the 14th day of the intragastric administration of compound *Lactobacillus*, eyeball blood was collected from mice in the two experimental groups, and 1 mL of blood was put into a centrifuge tube for centrifugation. The serum was separated, and the total antioxidant capacity of the mouse serum was detected according to the instructions of the mouse total antioxidant capacity (T-AOC) enzyme-linked immunosorbent assay kit (Enzyme-free Biotechnology Co., Ltd., Chang Chun, China).

### 2.9. Inhibition Testing of Lactobacillus Combination Preparation against C. rodentium In Vivo

On the 15th day of the experiment, the mice in the PBS group and compound *Lactobacillus* group began to be intragastrically administered 200 µL of *C. rodentium* at a concentration of 1 × 10^9^ CFU/mL for two days. After the challenge, the weight changes of mice were recorded every other day for 14 consecutive days, and charts were made. The feces of the mice in the two groups were collected every other day and normalized to the same weight. The treatment of feces is to dissolve it with sterile water to make a feces suspension. Then, 100 μL of the feces suspension was mixed with 900 μL of sterile water, labeled D1. Then, 100 μL of feces suspension was drawn and mixed with 900 μL of sterile water, labeled D1, and 100 μL from D1 and mixed with 900 μL of sterile water, labeled D2, and so on. An amount of 100 μL from D1 was then mixed with 900 μL of sterile water, labeled D2, and so on, finally labeled D10. Then, 100 μL of the D7, D8, D9 and D10 liquid were inoculated into citrobacter chromogenic medium, finally labeled D10. Finally, 100 μL of D7, D8, D9 and D10 were inoculated into Citrobacter chromogenic medium, respectively. After culturing for 12 h, the colony numbers were counted. Among them, citrobacter chromogenic medium is a specific medium—only the citrobacter grew again—other bacteria did not grow, and the change in colony number was recorded for 14 consecutive days.

### 2.10. Changes in IFN-γ IL-10 Levels Detected by ELISA

On the 14th day after the challenge, the eyeball blood was collected from the eyes of mice in the two experimental groups. One milliliter of blood was put into a centrifugal tube for centrifugation, and the serum was separated. The levels of IFN-γ and IL-10 were determined according to the instructions of enzyme-linked immunoassay kits (Enzyme-Free Biology Company, Shang Hai, China).

### 2.11. Histomorphology

The colon and duodenum of the mice were fixed with 10% formalin, and the intestinal tissue of appropriate size was put into an embedding box, placed in 70% ethanol, transferred from 70% ethanol to high-concentration ethanol for dehydration, and then made transparent and dipped in wax. The intestinal tissue was vertically embedded in paraffin and then sectioned, spread, baked and stained with haematoxylin-eosin.

### 2.12. Flow Cytometry

On day 30 of the experiment, three mice in each group were euthanized. The spleen, mesenteric lymph nodes (MLNs) and Pyle’s plaques (PPs) were removed aseptically to prepare single-cell suspensions. The prepared 100 mL PPs cell suspension (1 × 10^6^ cells) was mixed with anti-CD16 /CD32 (BD Biosciences, San Jose, CA, USA) at 4 °C and incubated for 5 min. Anti-CD3-AF700 (BD Biosciences, San Jose, CA, USA), anti-CD4-PECy7 (BD Biosciences, San Jose, CA, USA), and anti-CD8-APC (BD Biosciences, San Jose, CA, USA) were then added directly. The antibodies were incubated at 4 °C in the dark for 20 min; then, the cell suspension was washed. The samples were examined by flow cytometry (BD SRFortessa™, San Jose, CA, USA) through a nylon sieve. All data were analyzed using FlowJo 7.6.2 software (Tree Star, Inc., San Jose, CA, USA).

### 2.13. Intestinal Microbiota Sequencing

The 341F (5′-CCTAYGGGRBGCASCAG-3′) and 806R (5′GGACTACNNGGGTATCTAAT-3′) primers were used to amplify the V3–V4 hypervariable region of 16S rRNA gene in intestinal bacteria. For Illumina NovaSeq analysis, a small fragment library was constructed and sequenced by Paired_End based on the Illumina NovaSeq sequencing platform. After reading splicing and filtering, operational taxon (OTU) clustering, species annotation and abundance analysis, and deep data mining based on α diversity and β diversity analysis were performed [6,7].

### 2.14. Statistical Analysis

The experimental data were statistically analyzed with GraphPad Prism 5.0 software (GraphPad Software Inc., San Diego, CA, USA) and expressed as the mean ± SD. * *p* < 0.05, ** *p* < 0.01, and *** *p* < 0.001 compared to controls were calculated by one-way ANOVA with a post-hoc Tukey’s test.

## 3. Results

### 3.1. Acid Production and Bacteriostatic Activity of Lactobacillus Combination Preparation

The acid production and bacteriostatic activity of the *Lactobacillus* combination are shown in Figure 1. According to the results in the figure, in the test, after the combination of *Lactobacillus lactis* L9 and *Enterococcus faecalis* L16 according to different combination ratios, the pH values of fermentation broth of the five groups were only 1:1, 1:2, 1:3, 1:4 and 4:1, which were lower than that with the single strain. When the ratio was 4:1, the pH was the lowest (Figure 1A), and the inhibition zone was the largest, reaching 16.69±0.04 mm. The inhibition diameters of each compound *Lactobacillus* group were all larger than that of strain L9 alone, and the bacteriostatic diameters of each group were all larger than that of the ultrapure water group at pH 4.0 (Figure 1B). The results showed that the pH value was the lowest and the bacteriostatic diameter was the largest when the compound ratio of *Lactobacillus lactis* L9 to *Enterococcus faecalis* L16 was 4:1, which proved that the optimal compound ratio of *Lactobacillus lactis* L9 to *Enterococcus faecalis* L16 was 4:1.

### 3.2. Effects of the Lactobacillus Combination on the Body Weight of Mice

The influence of the *Lactobacillus* combination on the body weight of mice is shown in Figure 2. With the increase in days of intragastric administration, the weight gain of mice in the compound *Lactobacillus* group was greater than that in the PBS group. The average daily gain of the PBS group was 0.09 g, and the average daily gain of the compound *Lactobacillus* group was 0.13 g. The results indicated that the body weight of mice could be increased by feeding the lactic acid bacteria combination.

### 3.3. Effects of the Lactobacillus Combination on Feed Intake in Mice

The effect of the *Lactobacillus* combination on the feed intake of mice is shown in Figure 3. The average daily feed intake of the compound *Lactobacillus* and PBS groups was 1.20 g and 1.19 g, respectively. The feed intake of the mice fed the compound probiotics was slightly lower than that of mice fed PBS, indicating that the compound probiotics could slightly reduce the feed intake of mice.

### 3.4. The Ratio of Feed to Gain

The feed/gain ratio of mice in the PBS group was 13.33, and the feed/gain ratio of mice in the compound *Lactobacillus* group was 9.15, indicating that the *Lactobacillus* combination could increase the weight of mice, reduce the feed intake of mice, reduce the feed/gain ratio, increase economic utility, and have a good probiotic effect.

### 3.5. Serum Total Antioxidant Capacity Detected by ELISA

The total antioxidant capacity in the serum of mice was measured according to the instructions of the enzyme-linked immunosorbent assay kit. The results are shown in Figure 4. The total antioxidant capacity in the serum of mice after gastric administration of *Lactobacillus* was higher than that of mice in the PBS group (* *p* < 0.05).

### 3.6. In Vivo Bacteriostatic Activity Experiment

#### 3.6.1. Effects of the Lactobacillus Combination on the Body Weight of Mice after Challenge

The influence of the *Lactobacillus* combination on the body weight of mice after the challenge is shown in Figure 5. The results showed that the body weight of mice in the PBS group decreased from Day 1 to Day 6 and then increased slowly for 14 days after the *C. rodentium* challenge. The body weight of mice in the compound *Lactobacillus* group decreased slightly from Day 1 to Day 4 after the *C. rodentium* challenge and then showed a trend of slow growth.

#### 3.6.2. Changes in the Number of *C. rodentium* in Mouse Faeces

As shown in Figure 6, the number of *C. rodentium* in the PBS group was greater than that in the compound *Lactobacillus* group. The maximum value of the number of *C. rodentium* in the PBS group was reached on the 6th day of intragastric administration, the maximum value of the number of *C. rodentium* in the compound *Lactobacillus* group was reached on the 4th day of the challenge, and the results corresponded to the change in body weight.

#### 3.6.3. Changes in the Levels of the Cytokines IFN-γ and IL-10 Were Detected by ELISA

The changes in the levels of the cytokines IFN-γ and IL-10 are shown in Figure 7. The serum IFN-γ content in the PBS group was higher than that in the compound *Lactobacillus* group (Figure 7A), and the IL-10 content in the compound *Lactobacillus* group was higher than that in the PBS group (Figure 7B). This result indicated that the *Lactobacillus* combination can decrease serum IFN-γ levels and increase serum IL-10 levels.

#### 3.6.4. Histomorphological Changes in the Mouse Colon and Duodenum

Histopathological changes in the duodenum and colon of mice in the PBS group, combined *Lactobacillus* group and blank group are shown in Figure 8. The results showed that there were obvious inflammatory symptoms in the colonic tissues of mice in the PBS group (Figure 8A), including a large number of lymphocytes, neutrophil infiltration, varying numbers of eosinophils, and reduction in the number of goblet cells. The colons of mice in the compound *Lactobacillus* group had low inflammatory injury, which manifested as slight inflammatory cell infiltration (Figure 8B). No obvious histopathological damage was observed in the blank group (Figure 8C). This result suggests that the *Lactobacillus* combination may reduce intestinal inflammation.

#### 3.6.5. Changes of T Lymphocyte Number in Mice

After observation, T lymphocytes in the spleen and MLN of mice in each group were detected by flow cytometry. The results showed that compared with the healthy group, compound lactic acid bacteria significantly increased the number of CD4^+^ and CD8^+^T lymphocytes in spleen T lymphocytes and enhanced the cellular immune response (Figure 9A,B). At the same time, the compound *Lactobacillus* significantly increased the number of CD4^+^T lymphocytes in the T lymphocytes of MLN. The T lymphoid quantity of the compound lactic acid bacteria group was higher than that of the PBS group (Figure 9C,D).

#### 3.6.6. Effects of Compound Lactic Acid Bacteria on Intestinal Microbiota of Mice

The composition and changes of bacterial microbiota in the feces of mice before and after feeding compound lactic acid bacteria were detected. First, the α diversity results showed that mice fed with the compound *Lactobacillus* increased their α diversity and increased their species richness after being infected with *C. rodentium*. (Table 1). Non-metric multidimensional scaling (NMDS) was used for generic-level clustering of samples. The results showed that there were significant differences in microbiota distribution between the mice fed complex lactic acid bacteria and the control group after infection with *C. rodentium*. (Figure 10A). When infected with C. *rodentium*, the species composition of mice fed with the *Lactobacillus* complex increased. (Figure 10B). During infection with *C. rodentium*, the species composition of mice fed complex lactic acid bacteria increased. Among them, the relative abundance of *Muribaculaceae* and *Alistipes* increased (Figure 10C). At the genus level, the relative abundance of *Alistipe*, *Odoribacter* and *Eubacterium xylanophilum Group* increased, and the relative abundance of *Absiella* and *Ligilactobacillus* decreased (Figure 10D). The effects of *C. rodentium* on intestinal microbiota function were predicted before and after infection with *C. Rodentium* in mice fed with the *Lactobacillus* complex. The results showed that the General function prediction only, Porphyrin and pattern and energy metabolism in leaves significantly increased in mice fed the compound lactic acid bacteria. (Figure 11).

## 4. Discussion

Studies have shown that Lactobacillus can inhibit the growth of some pathogenic microorganisms by producing organic acids and even killing pathogenic microorganisms, but it is not the only way [8,9,10]. Piglet enteritis caused by *Escherichia coli* is a common disease in the pig industry [11,12]. *Escherichia coli* adheres to host intestinal epithelial cells through adhesins, releases enterotoxins during intestinal reproduction, and produces toxic effects on intestinal epithelial cells, eventually causing enteritis, diarrhea, dehydration and even death [13,14]. Based on this background, this study included the examination of the acid production capacity of *Lactobacillus* at different ratios and the inhibition of *Escherichia coli* as the screening conditions for different compound ratios. The results show that the 4:1 ratio of *Lactobacillus* L9:L16 produces the strongest acid capacity. Similarly, when the *Lactobacillus* L9:L16 ratio was 4:1, the bacteriostatic diameter was the largest. The best ratio of strains L9 and L16 of 4:1 was determined to achieve the best acid production and bacteriostatic effect.

*Lactobacillus* preparations can accelerate the growth rate of animals [15]. The organic acids produced by *Lactobacillus* in the process of reproduction can stimulate the conversion of pepsinogen into pepsin, accelerate the absorption of protein, accelerate intestinal peristalsis and increase the growth of animals [16,17]. In this experiment, after intragastric administration of the *Lactobacillus* combination to mice, it was found that the weight gain rate of mice in the compound *Lactobacillus* group was significantly higher than that in the PBS group, which effectively reduced the ratio of feed to gain, proving that the *Lactobacillus* combination had certain growth-promoting ability.

In animals, during the metabolism of aerobic cells, unstable, highly active oxygen substances will be produced, which will damage the health of animals. In this experiment, the total antioxidant capacity in serum was detected by ELISA. It was found that the *Lactobacillus* combination can improve the total antioxidant capacity of serum to reduce the production of reactive oxygen species and enhance immunity. This finding is similar to the results of Xu et al. [18], which showed that probiotics can significantly improve daily gain, antioxidant capacity and intestinal immune function of piglets and that early addition of probiotics can promote the healthy growth of piglets.

Studies have shown that *C. rodentium* has the same pathogenic gene and pathogenic mechanism similar to that of enteropathogenic *Escherichia coli* and enterohemorrhagic *Escherichia coli* [19]. Moreover, mice are not susceptible to *E. coli* infection. Therefore, *C. rodentium* infection has become a model for inducing colitis in mice [20]. In this experiment, mice were intragastrically fed *C. rodentium* to study the inhibitory effect of the strain combination on *C. rodentium*-induced colitis in mice in vivo. The Lactobacillus complex colonized in mice. After infection with Citrobacter, the colonized Lactobacillus complex could alleviate the inflammation of the mouse colon caused by Citrobacter. The results showed that the body weight of mice fed with the *Lactobacillus* combination in advance decreased slightly in the first 4 days and then continued to increase, while the body weight of mice in the PBS group decreased to the lowest point on the 6th day and then increased slowly. The decrease was greater than that in compound *Lactobacillus* group. Every two days, the feces of mice were inoculated onto MacConkey medium, and colonies were counted to observe the number of *C. rodentium*; the results showed that the number of *C. rodentium* in the compound *Lactobacillus* group was less than that in the PBS group. The results of the two experiments showed that the *Lactobacillus* combination could inhibit *C. rodentium* proliferation in the intestinal tract, and the addition of *Lactobacillus* to piglet feed could prevent *Escherichia coli*-induced diarrhea.

Studies [21,22,23] have shown that *Lactobacillus* can protect the host from pathogens by regulating the balance of proinflammatory (IL-6, IL-12, TNF, and IFN-γ) and anti-inflammatory (IL-10) cytokines. *C. rodentium* can cause T lymphocytes to infiltrate into the mucosal lamina propria in mice and increase the expression of IFN-γ, IL-12 and TNF-α in mice [24]. These cytokines can participate in the protective immunity of the host. Elliott et al. reported that removal of IL-12 and TNF-α receptors reduced the resistance of mice to *C. rodentium* infection. In this experiment, ELISA kits for IFN-γ and IL-10 were used to assay the serum of mice after the challenge [22,25]. The results showed that the contents of the proinflammatory cytokines IFN-γ in the *Lactobacillus* group were lower than those in the PBS group, while the content of the anti-inflammatory cytokine IL-10 was higher than that in the PBS group. The results indicated that the strain combination could reduce the content of proinflammatory factors and increase the content of anti-inflammatory factors in vivo, which could improve the immunity of the body.

The colon of mice after *Citrobacter* intragastric administration showed obvious inflammatory symptoms, including thickening of the distal colon wall, hyperplasia of fibrous tissue and hyperaemia. Inflammatory cells infiltrated the lamina propria of the mucosa, goblet cell numbers decreased, and the intestinal gland around the inflammatory site was disordered and damaged. The *Lactobacillus* combination can reduce intestinal mucosal damage and inflammation.

*C. rodentium* infection affects the gut microbiota of mice. Mice fed with *Lactobacillus* complex can resist *C. rodentium* infection and stabilize the structure of gut microbiota. However, mice fed with compound *Lactobacillus* could resist *C. rodentium* infection and stabilize the structural changes of gut microbiota. In this experiment, the relative abundance of Muribaculaceae_unclassified, Lachnospiraceae_NK4A136_group, Alistipes, Clostridiales and other beneficial bacteria in the intestinal tract of mice increased significantly after feeding compound lactic acid bacteria. However, the relative abundance of harmful bacteria such as Bacteroides was significantly reduced, and the function of Energy metabolism of gut microbiota was enhanced, this result is similar to that of Seo Kun-ho [6].

In conclusion, the *Lactobacillus* combination can promote animal growth, improve the feed conversion rate, and maintain the balance of the intestinal microbiota. At the same time, the *Lactobacillus* combination could inhibit colitis induced by *C. rodentium* in mice.

## Figures and Tables

**Figure 1 microorganisms-12-00730-f001:**
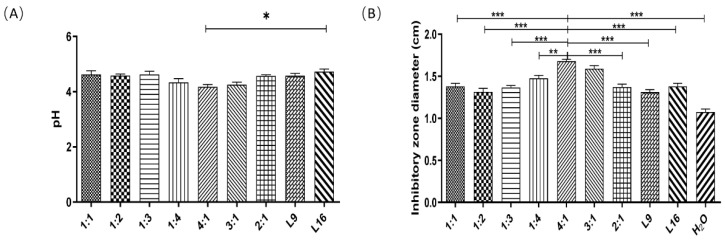
Acid production and bacteriostatic activity of the Lactobacillus combination. 1:1, 1:2, 1:3, 1:4, 4:1, 3:1, and 2:1 in the figure indicate the ratios of L9 to L16. The compound acid production (**A**) and compound bacteriostatic activity (**B**). Data are represented as the mean ± SEM. Statistically significant differences are indicated (* *p* <0.05; ** *p* < 0.01; *** *p* < 0.001), and the line above the column marks the two groups with differences.

**Figure 2 microorganisms-12-00730-f002:**
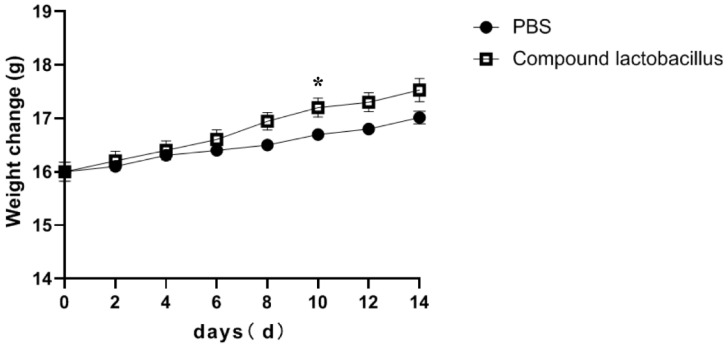
Changes in the body weight of mice after treatment (* *p* < 0.05).

**Figure 3 microorganisms-12-00730-f003:**
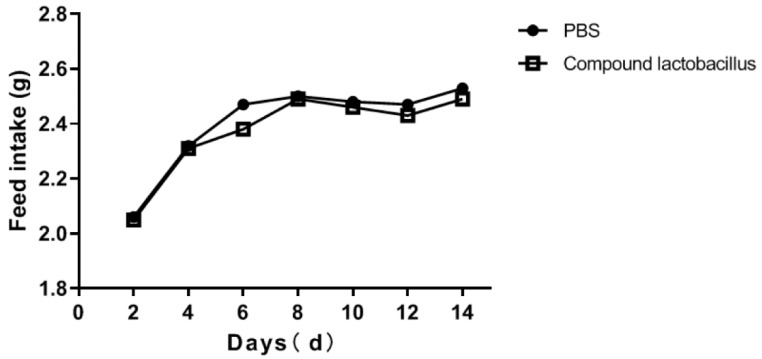
Changes in the feed intake of mice after treatment.

**Figure 4 microorganisms-12-00730-f004:**
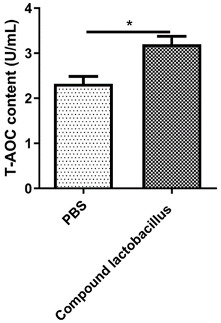
Total antioxidant capacity in serum. Data are presented as the mean ± SEM of two experiments (*n* = 12 mice per group) (* *p* < 0.05).

**Figure 5 microorganisms-12-00730-f005:**
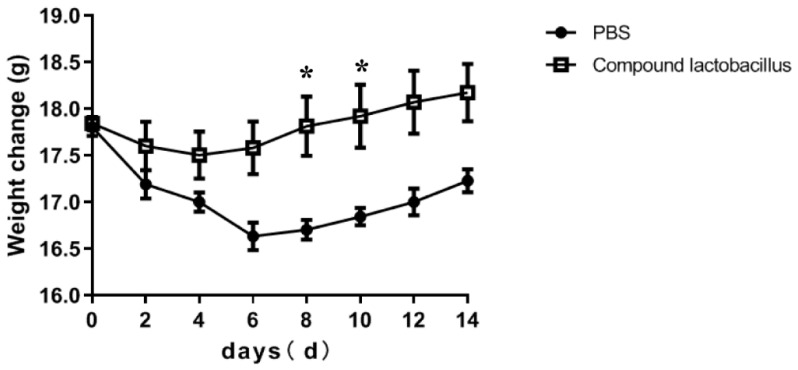
Effects of the *Lactobacillus* combination on the body weight of mice after challenge (* *p* < 0.05).

**Figure 6 microorganisms-12-00730-f006:**
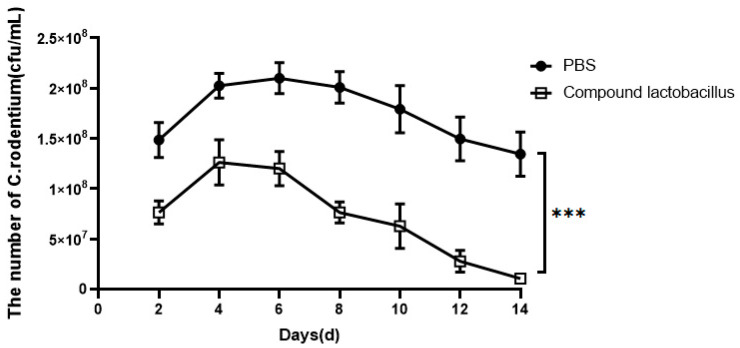
Changes in the number of *C. rodentium* in faeces (*** *p* < 0.001).

**Figure 7 microorganisms-12-00730-f007:**
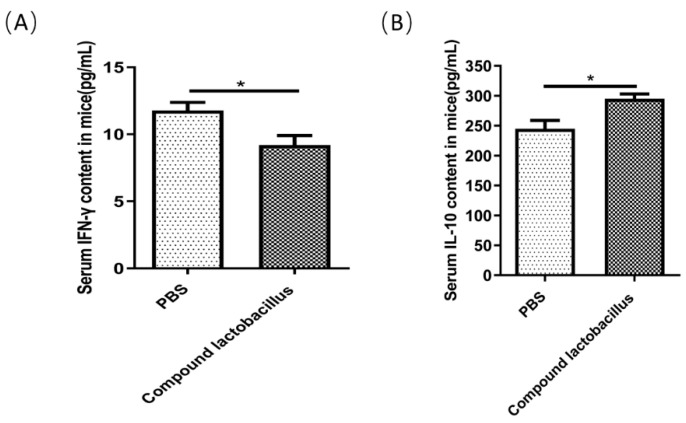
Changes in the levels of the cytokines IFN-γ and IL-10. The content of IFN-γ in the serum of mice (**A**) and the content of IL-10 in the serum of mice (**B**). Data are presented as the mean ± SEM of two experiments (*n* = 12 mice per group) (* *p* < 0.05).

**Figure 8 microorganisms-12-00730-f008:**
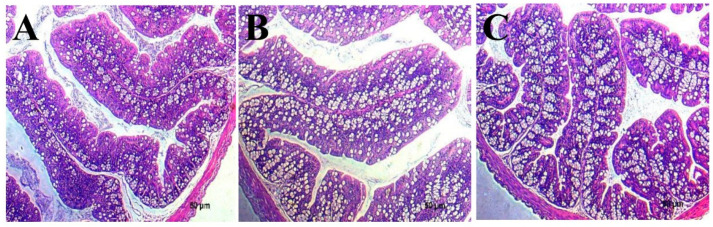
The histomorphological changes in the mouse colon and duodenum. Representative haematoxylin and eosin-stained sections of the colon (magnification: 400×). Pathological sections of the colon of mice in the PBS group (**A**), compound Lactobacillus group (**B**), and blank group (**C**).

**Figure 9 microorganisms-12-00730-f009:**
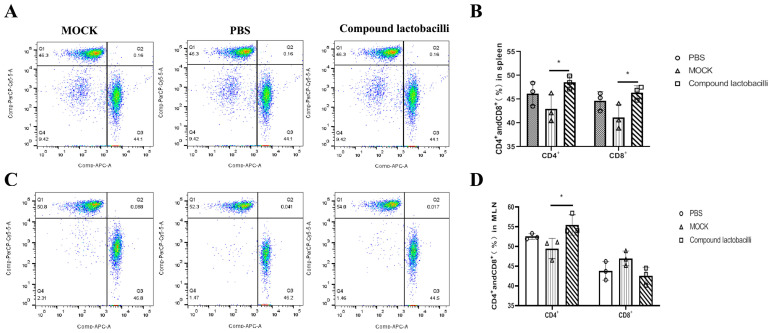
T lymphocytes in the spleen and MLN were detected by flow cytometry. Results of flow cytometry in the spleen of mice in each group (**A**), statistical analysis of CD4^+^T lymphocytes and CD8^+^T lymphocytes in the spleen of mice in each group (**B**), MLN flow cytometry detection results of mice in each group (**C**), and statistical analysis of CD4^+^T lymphocytes and CD8^+^T lymphocytes in MLN of mice in each group (**D**). Data are presented as the mean ± SEM of two experiments (*n* = 3 mice per group) (* *p* < 0.05).

**Figure 10 microorganisms-12-00730-f010:**
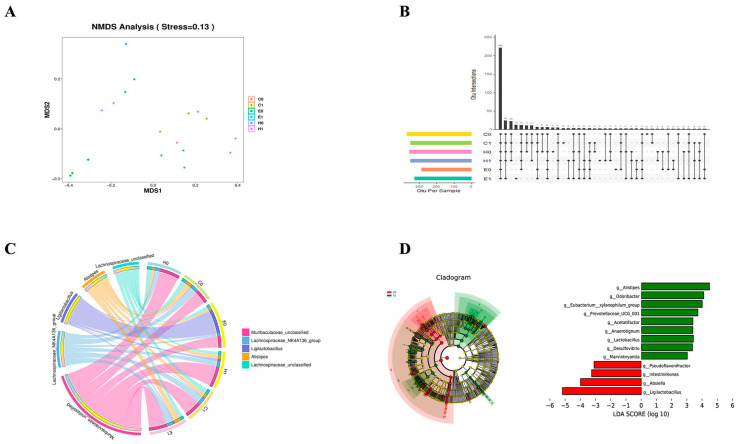
S-sequencing data of intestinal microflora of the mice in each group. Beta diversity analysis (**A**), species diversity analysis (**B**), analysis of the relative abundance of species (**C**), and LEfSe analysis at the genus level (**D**).

**Figure 11 microorganisms-12-00730-f011:**
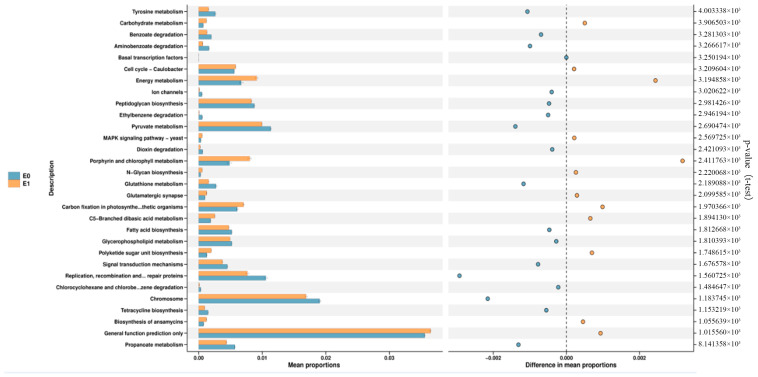
The function of intestinal flora in mice before and after infection with *C. rodentium* was compared.

**Table 1 microorganisms-12-00730-t001:** Comparison of α diversity in each group.

	Observed	Shannon	Chao1
E0	679	6.028017	679.308
E1	953.3333333	7.801402	953.3898
H0	1070	7.999735	1070.657
H1	808.6666667	7.533267	808.9767
C0	996.3333333	7.852373	996.7747
C1	1035.333333	8.070176	1035.864

## Data Availability

The raw amplicon sequencing data acquired in this study have been deposited at the China National Center for Bioinformation under the accession code CRA007544. The authors declare that all other data supporting the findings of the study are available in the paper and supplementary materials, or from the corresponding author(s) upon request. The raw data reported in this manuscript have been deposited in Jilin Agricultural University, Changchun, China.

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
