# Peer review of "Inhibitory Effect of Lactococcus and Enterococcus faecalis on Citrobacter Colitis in Mice"

_microorganisms, 2024, doi:10.3390/microorganisms12040730_

Round 1
Reviewer 1 Report
Comments and Suggestions for Authors
The aim of the manuscript was to investigate the inhibitory effect of the combination of Lactobacillus and Enterococcus faecalis isolates on an experimental model of colitis induced by Citrobacter rodentium. Furthermore, the capacity to promote growth, feed conversion rate, and maintain the balance of the intestinal microbiota in mice was evaluated. It was found that the combination of Lactobacillus can promote animal growth, improve the feed conversion rate, maintain the balance of intestinal microbiota, and inhibit C. rodentium-induced colitis in mice. The text is unclear and lacks reference citations in the Materials and Methods section. Overall, the manuscript must be entirely revised. The comments were as follows:
- Abstract and other parts of the manuscript: Scientific names are italicized.
- In many parts of the text the authors use the term "flora, microflora and microbiota": please, preferably use the term "microbiota";
- Line 56-57: In the sentence "Then, the dominant Lactobacillus strains L9 and L19 were 56 mixed at a specific compound ratio" it is not clear why the expression "dominant" is used. I suggest that this be changed.
- Line 66-73: The sentence states that bacterial identification was carried out by 16S rDNA sequencing. In this case, what is the species name for the Lactobacillus L9 isolate? This is important for studies of this nature. Furthermore, the isolates were obtained from piglet feces and the experiments were carried out in mice. Intestinal colonization is necessary to act as probiotics. Is there evidence that this could have occurred with certainty? Please clarify the strain number of the C. rodentium used.
- Lines 83-88: The text is very confusing. How precisely was this combination of bacteria accomplished? The authors mentioned a third species (Pediococcus lactis L9) in combination with Enterococcus faecalis.
Lines 91-96: The paragraph is also confusing and some terms do not correspond to what is used in microbiology, such as "bacterial solution" (bacterial suspension); "the straight diameter of the bacteriostatic circle was measured" (the diameter of the inhibition zone was measured). Furthermore, the description of how the method was performed is not understandable - "Three holes were drilled with a 6 mm hole punch, and 100 μl of ultrapure water at pH 4.0, each compound bacterial solution and two single bacterial solutions were transferred to the three holes" - implies that there were three cavities in the agar, but that four items were added to the wells (?). In this paragraph, there is another question: would an incubation of just 8h be enough to obtain a reliable inhibition zone of growth (please cite the reference)?
Lines 140-142: Please explain the dilution of the faeces (i.e., decimal serial dilutions?). After inoculating the dilutions onto MacConkey agar, incubation was carried out for 12h - would this time be sufficient to reveal all viable Citrobacter colonies. How were Citrobacter colonies differentiated from other non-fermenting lactose bacteria?
- Lines 167-168: please cite the reference of the primers used to determine the microbiome.
- Line 185: I believe the authors meant "mm" and not "cm";
- Lines 203-203: the authors stated that the Lactobacillus combination performed well as a growth promoter; please indicate whether the difference was statistically significant.
- Lines 216-219: please indicate whether the difference is significant.
- Line 246, figure 6: Please indicate the p-value.
Line 293: Please, non-metric is misspelled as non-netric.
- Line 319: The production of organic acids by Lactobacillus is not the only mechanism for inhibiting the growth of other bacteria.
- Line 357 (also 142): Please use "the mouse feces were inoculated onto MacConkey medium" instead of "coated on ..."
Comments on the Quality of English LanguageThe manuscript must necessarily undergo English revision to make the language clearer and more concise
Author Response
- Abstract and other parts of the manuscript: Scientific names are italicized.
It has been modified according to your comments
- In many parts of the text the authors use the term "flora, microflora and microbiota": please, preferably use the term "microbiota";
It has been modified according to your comments
- Line 56-57: In the sentence "Then, the dominant Lactobacillus strains L9 and L19 were 56 mixed at a specific compound ratio" it is not clear why the expression "dominant" is used. I suggest that this be changed.
It has been modified according to your comments
- Line 66-73: The sentence states that bacterial identification was carried out by 16S rDNA sequencing. In this case, what is the species name for the Lactobacillus L9 isolate? This is important for studies of this nature. Furthermore, the isolates were obtained from piglet feces and the experiments were carried out in mice. Intestinal colonization is necessary to act as probiotics. Is there evidence that this could have occurred with certainty? Please clarify the strain number of the C. rodentium used.
It has been modified according to your comments
- Lines 83-88: The text is very confusing. How precisely was this combination of bacteria accomplished? The authors mentioned a third species (Pediococcus lactis L9) in combination with Enterococcus faecalis.
Hello, we are studying animal microecological preparations, that is, combining animal-derived lactic acid bacteria in accordance with the optimal proportion, and then exploring the effects of the composite microecological preparations through in vitro and in vivo experiments. Therefore, in this paper, we adopted 1:1, 1:2, 1:3, 1:4, 2:1, 3:1, 4:1 and other ratios, and combined the number of lactic acid bacteria according to this ratio. The ratio of lactic acid bacteria will also affect the effect produced after the combination, and there will be antagonism between lactic acid bacteria.
Lines 91-96: The paragraph is also confusing and some terms do not correspond to what is used in microbiology, such as "bacterial solution" (bacterial suspension); "the straight diameter of the bacteriostatic circle was measured" (the diameter of the inhibition zone was measured). Furthermore, the description of how the method was performed is not understandable - "Three holes were drilled with a 6 mm hole punch, and 100 μl of ultrapure water at pH 4.0, each compound bacterial solution and two single bacterial solutions were transferred to the three holes" - implies that there were three cavities in the agar, but that four items were added to the wells (?). In this paragraph, there is another question: would an incubation of just 8h be enough to obtain a reliable inhibition zone of growth (please cite the reference)?
I am very sorry for the confusion caused to you. I am deeply sorry for my language expression problem, and I will explain it. What we did was a bacteriinhibitory experiment. First, pathogenic bacteria were coated on the surface of LB's solid medium, and then a hole punch was used to make 4 holes in the solid medium, adding sterile water into the first hole as the control hole, and adding L9 into the second hole. L16 was added to the third well, and L9+L16 was added to the fourth well in accordance with the ratio of complex L9+L16(there were 7 groups according to different compounding ratios), and the size of antibacterial ring was observed after culture for 8h.
Lines 140-142: Please explain the dilution of the faeces (i.e., decimal serial dilutions?). After inoculating the dilutions onto MacConkey agar, incubation was carried out for 12h - would this time be sufficient to reveal all viable Citrobacter colonies. How were Citrobacter colonies differentiated from other non-fermenting lactose bacteria?
I am very sorry for the confusion caused to you. I am deeply sorry for my language expression problem, and I will explain it. The treatment of feces is to dissolve feces with sterile water and make feces suspension, then draw 100μl of feces suspension and mix with 900μl of sterile water, labeled D1, take 100μl from D1 and mix with 900μl of sterile water, labeled D2, and so on, finally labeled D10. Liquid 100μl of D7, D8, D9 and D10 were inoculated into MacConkey medium and citrobacter chromogenic medium, respectively. After culture for 12h, colony numbers were counted. Among them, McConkey's medium and citrobacter chromogenic medium are specific medium, only citrobacter grows again, and other bacteria do not grow.
- Lines 167-168: please cite the reference of the primers used to determine the microbiome.
All right, the references have been inserted
- Line 185: I believe the authors meant "mm" and not "cm";
Ok, it has been modified
- Lines 203-203: the authors stated that the Lactobacillus combination performed well as a growth promoter; please indicate whether the difference was statistically significant.
Ok, it has been modified
- Lines 216-219: please indicate whether the difference is significant.
Ok, it has been modified
- Line 246, figure 6: Please indicate the p-value.
Ok, it has been modified
Line 293: Please, non-metric is misspelled as non-netric.
Ok, it has been modified
- Line 319: The production of organic acids by Lactobacillus is not the only mechanism for inhibiting the growth of other bacteria.
What you said is correct. I will revise it
- Line 357 (also 142): Please use "the mouse feces were inoculated onto MacConkey medium" instead of "coated on ..."
Ok, it has been modified
Reviewer 2 Report
Comments and Suggestions for Authors
In the present experimental study, Naveed et al found that a combination 4:1 of two lactobacilli strains had the best effect on immunitary cell profile and microbiota composition in a murine model of colitis induced by Citrobacter. Main comments:
1) Please check sentence at page 1 lines 16-18, as I assume that lactobacilli were isolated from healthy piglets.
2) How is it possible to measure mental state of mice?
3) I do not understand the rationale of the experiment described in page 3 lines 119-123: what is the utility of measuring piglet feed intake?
4) Figures have no caption.
5) Figures 2-3-5-6 have no statistical test to support what described in the Results section.
6) Lines 245-254: this statement was not supported in any way, as inflammation was not scored and compared but evaluated only qualitatively and grossly.
7) As lactobacilli were administered before the Citrobacter, Authors can only assume that they may have a protective effect. Therefore Discussion should be rewritten based on such assumption.
Comments on the Quality of English LanguageNone
Author Response
In the present experimental study, Naveed et al found that a combination 4:1 of two lactobacilli strains had the best effect on immunitary cell profile and microbiota composition in a murine model of colitis induced by Citrobacter. Main comments:
1) Please check sentence at page 1 lines 16-18, as I assume that lactobacilli were isolated from healthy piglets.
Hello, forgive me for my unclear expression. I would like to explain that we are a research department engaged in animal microecological preparations, mainly dealing with animal microecological preparations for pigs. Therefore, we have isolated a lot of probiotics from pigs. I have modified the part you mentioned
2) How is it possible to measure mental state of mice?
By seeing if the rat's fur is smooth? See if the rats congregate? See if the mice eat? Can indirectly judge the mental state of mice.
3) I do not understand the rationale of the experiment described in page 3 lines 119-123: what is the utility of measuring piglet feed intake?
Firstly, feed intake can be used to determine whether the mice are hungry and whether the complex probiotics affect the digestion of the mice. Secondly, weight measurement can indirectly compare whether the complex probiotics have the function of promoting the growth of mice, which is very important in clinical production.
4) Figures have no caption.
Thank you for your valuable comments, which have been added at the end of the article
5) Figures 2-3-5-6 have no statistical test to support what described in the Results section.
Relevant changes have been made
6) Lines 245-254: this statement was not supported in any way, as inflammation was not scored and compared but evaluated only qualitatively and grossly.
Thank you very much for your valuable advice, and I agree with you very much. The article also made a simple evaluation on the relief of inflammation by complex lactic acid bacteria.
7) As lactobacilli were administered before the Citrobacter, Authors can only assume that they may have a protective effect. Therefore Discussion should be rewritten based on such assumption.
Thank you very much for your suggestions, which have been modified according to your suggestions.
Round 2
Reviewer 1 Report
Comments and Suggestions for Authors
The authors have accepted most of the suggestions, but the manuscript still contains modifications to be made before it can be accepted for publication. Additionally, the text of the manuscript must be proofread by a native English speaker for grammar and English usage.
Line 60: Please, remove one “then”;
Line 149-155 – The paragraph contains English errors and should be revised. Furthermore, the authors mistakenly stated that MacConkey agar is a specific culture medium for Citrobacter. Other members of the Enterobacteriaceae family (such as Escherichia coli, Klebsiella, Enterobacter, Morganella, Proteus, etc.), as well as non-fermenting Gram-negative bacilli (Pseudomonas, Acinetobacter, Burkholderia, etc.) also grow in this culture medium.
Line 315: The genus name Ligilactobacillus has been misspelled.
Comments on the Quality of English LanguagePlease, the manuscript must be proofread by a native English speaker for grammar and English usage.
Author Response
Dear expert, I have accepted your suggestion and modified it according to your suggestion. Thank you very much.
Line 60: Please, remove one “then”;
Have been deleted
Line 149-155 – The paragraph contains English errors and should be revised. Furthermore, the authors mistakenly stated that MacConkey agar is a specific culture medium for Citrobacter. Other members of the Enterobacteriaceae family (such as Escherichia coli, Klebsiella, Enterobacter, Morganella, Proteus, etc.), as well as non-fermenting Gram-negative bacilli (Pseudomonas, Acinetobacter, Burkholderia, etc.) also grow in this culture medium.
Thank you for your question. The screening of McConkey's culture-medium has been removed,
Line 315: The genus name Ligilactobacillus has been misspelled.
Modifications have been made,
Reviewer 2 Report
Comments and Suggestions for Authors
Regarding pont 2, these evaluations are not effective for mental status. Therefore please remove fro, the article data about it.
Regarding point 6, I do not see any satisfactory answer. An inflammation score and a statistical analysis is still missing.
Comments on the Quality of English LanguageNone
Author Response
Thank you for your advice. I have realized my mistake and have modified it. IL-10 is an anti-inflammatory factor, while IFN-γ is a pro-inflammatory factor, both of which are indicators for evaluating inflammation.
Regarding pont 2, these evaluations are not effective for mental status. Therefore please remove fro, the article data about it.
Thank you for your suggestion. It has been deleted according to your suggestion
Regarding point 6, I do not see any satisfactory answer. An inflammation score and a statistical analysis is still missing.
Thank you for your suggestion. The content of the article has been modified according to your suggestion.